# Effectiveness of Social Inclusion Interventions for Anxiety and Depression among Adolescents: A Systematic Review

**DOI:** 10.3390/ijerph20031895

**Published:** 2023-01-19

**Authors:** Xanthe Hunt, Tom Shakespeare, Gabriele Vilyte, G. J. Melendez-Torres, Junita Henry, Melissa Bradshaw, Selvan Naidoo, Rachel Mbuyamba, Shahd Aljassem, Esta Suubi, Nawar Aljasem, Moroesi Makhetha, Jason Bantjes

**Affiliations:** 1Institute for Life Course Health Research, Department of Global Health, Stellenbosch University, Cape Town 7505, South Africa; 2International Centre for Evidence on Disability, London School of Hygiene and Tropical Medicine, London WC1E 7HT, UK; 3College of Medicine, University of Exeter, Exeter EX4 4QJ, UK; 4Alcohol, Tabaco and Other Drug Research Unit, South African Medical Research Unit, Cape Town 7505, South Africa

**Keywords:** social inclusion, active ingredients, systematic review, mental health, lived experience

## Abstract

Background: Adolescents who are socially excluded are at increased risk of mental health problems such as depression and anxiety. Promoting social inclusion could be an effective strategy for preventing and treating adolescent depression and anxiety. Methods: We conducted a systematic review of intervention studies which aimed to prevent or treat adolescent depression and/or anxiety by promoting social inclusion. Throughout the review we engaged a youth advisory group of 13 young people (aged 21–24) from Uganda, Turkey, Syria, South Africa, and Egypt. Results: We identified 12 studies relevant to our review. The interventions tested use a range of different strategies to increase social inclusion and reduce depression and anxiety, including social skills training, psychoeducation, teaching life skills training, and cash transfers. Pooled standardised mean differences (SMDs) based on random-effects models showed medium-to-large benefits of interventions on improving depression and anxiety symptoms (n = 8; SMD = −0.62; 95% CI, −1.23 to −0.01, *p* < 0.05). Conclusion: Although there are not many studies, those which have been done show promising results that strongly suggest that social inclusion could be an important component of programmes to promote adolescent mental health.

## 1. Introduction

Depressive and anxiety disorders are the most common mental health problems among adolescents [1,2], with a worldwide pooled lifetime prevalence of 2.6% (CI 95% 1.7–3.9) and 6.5% (CI 95% 4.7–9.1), respectively [1]. Depressive and anxiety disorders typically have their onset during adolescence [3,4] and can disrupt social, cognitive, and emotional development [5]. Left untreated, depression and anxiety are associated with a range of adverse outcomes, including suicide [6,7,8,9]. Significant advances have been made to identify evidence-based interventions for adolescent depression and anxiety [10,11,12], but most standard prevention and treatment strategies focus on intrapsychic (psychological phenomena that occur within the mind), cognitive, behavioural, and pharmaceutical interventions, with the implicit assumption that these conditions are mostly a function of internal processes [13,14,15,16,17,18]. There is, however, growing evidence that social determinants, including poverty, discrimination, and marginalisation, have a profoundly deleterious impact on mental health [19,20,21,22], including among adolescents [23,24,25]. Reducing discrimination and marginalisation by promoting social inclusion may thus be important for reducing adolescent depression and anxiety, particularly for adolescents who experience social exclusion, including, for instance, LQBTQ+ youth, refugees, and adolescents with disabilities [26,27,28].

The term “social inclusion” has become increasingly prominent in national policies and international development discourses. However, the term is contested and has been used to denote a variety of related constructs [29]. Social inclusion can be broadly defined as “improving the terms of participation in society for people who are disadvantaged on the basis of age, sex, disability, race, ethnicity, origin, religion, or economic or other status, through enhanced opportunities, access to resources, voice and respect for rights” [30] (p. 17). In contrast, social exclusion is defined as “a state in which individuals are unable to participate fully in economic, social, political and cultural life, as well as the process leading to and sustaining such a state” [30] (p. 180). The term “social inclusion” can be used as a verb to describe practices and processes that promote social integration and increase access to social capital. While social exclusion refers to processes and/or practices by which individuals are confined to the margins of society because they are members of a particular social group or because of a social identity or physical characteristic [31], social exclusion can be defined as the process through which individuals or groups are excluded (entirely or partially) from participating in society. Examples of acts of social exclusion include subjugating, disempowering or dispossessing people; prejudice, discrimination, and stigmatisation; violating human rights and preventing access to legal processes; restricting access to resources, healthcare, and education, and restricting movement or segregating people.

The term “social inclusion” is also used to describe what happens when people’s access to society increases. In contrast, the term “social exclusion” denotes the outcome of processes and practices that marginalise people, including social isolation, inequality, unemployment, poverty, and an inability to participate in the normal activities of citizens in society [32,33,34]. Social inclusion has a spatial component and can thus be enacted through structural and concrete changes that improve people’s geographic, social, and economic mobility, and access to spaces [35,36,37]. An obvious example of the spatial dimension of social inclusion is the universal design of buildings and physical spaces to make them accessible to people with disabilities. Social exclusion, on the other hand, results in restrictions on movement that confine people to physical spaces and particular roles or positions in society.

Social exclusion has several psycho-social consequences including depression and anxiety, relational problems, loss of identity, loss of cultural affiliations, de-integration from family ties, isolation, and deprivation [36]. Social exclusion contributes to poverty, lack of educational and employment opportunities, ill-health and disability, homelessness, poor social networks, lack of community participation, disparities, inequalities, and lack of personal safety [38].

There are groups of individuals who are marginalised in society through political and economic processes which render them more vulnerable to social exclusion, including LQBTQ^+^ individuals, persons with disabilities, ethnic and racial minorities, refugees, religious minorities, and people living in poverty. Some of the common themes and contributors to poor social inclusion that exist are the negative impact of poor social capital and a lack of social participation, a lack of education and unemployment, and poor housing in disadvantaged neighborhoods [39]. Recent research with people with lived experience of mental health conditions suggests that certain elements of social inclusion are particularly important; for example, those relating to social participation, social supports, housing, neighbourhood, community involvement, employment and education, health and well-being, and service utilisation [39].

While there is some general agreement in the literature on the processes by which people may become socially excluded, less attention has been paid to identifying mechanisms to promote social inclusion [38] (although some recent exceptions exist, see Gardner, et al. [40]).

A meta-analysis of studies on the relationship between perceived discrimination and mental health by Pascoe and Smart Richman [41] shows that experiences of social exclusion in the form of discrimination have a negative effect on depressive symptoms and general wellbeing, while social support and group identification, on the other hand, minimise these. Social inclusion may thus be an important pathway to promote positive psycho-social development, prevent symptoms of depression and anxiety, and promote recovery and relapse prevention among adolescents with depressive and anxiety disorders.

Social inclusion as a construct goes beyond social relatedness by accounting for the systemic and structural factors influencing relationships, by encompassing access to all social environments, and by focusing on political structures and practices that exclude individuals from society. It is, however, unclear what evidence there is to support social inclusion interventions to prevent or treat adolescent depression and anxiety (despite some evidence among adults).

We undertook a systematic review of evidence exploring the effectiveness of interventions to prevent or treat adolescent depression and/or anxiety by promoting social inclusion. Throughout the review process, a youth advisory group of 13 young people (aged 21–24) from Uganda, Turkey, Syria, South Africa, and Egypt were involved in this work. The involvement of experts with lived experienced is particularly important in research involving youth [42,43]. Their involvement in this review was critical to ensuring that the definition of concepts and framing and interpretation of findings resonated with the lived experience expertise of young people themselves.

Our review focused on answering the following questions:What types of interventions are being delivered to prevent or treat adolescent depression and/or anxiety by promoting social inclusion?How effective are these interventions?Are there specific groups of adolescents for whom these interventions are most effective?What are the mechanisms through which these interventions reduce adolescent depression and anxiety?

## 2. Materials and Methods

A review protocol was developed in accordance with the Preferred Reporting Items for Systematic Reviews and Meta-Analyses Protocol (PRISMA-P) guidelines through a participatory process (see Appendix A).

As noted above, throughout the review we engaged a youth advisory group of 13 young people. This group was engaged by the research team during key steps in the review process which required youth input. This consultative process has been used by the first author before [44], and ensures that the review and its findings include the reflections and expertise of young people. These contact points took the form of online consultations, where the research team presented and invited feedback on the proposed protocol and provided progress reports. For instance, during one consultation, the proposed protocol was presented to the youth advisory group, and they were asked for input about whether the definition of social inclusion was correct, and the kinds of outcomes were appropriate. There were three points of contact with the whole group during the project (over the course of 4 months), and one on one interactions with members of the group who were interested in contributing to the manuscript. Contact point one concerned a discussion of social inclusion and elicited feedback from the group on how best to define social inclusion, and how to frame its relevance in terms of their lives. The second contact point was a progress update from the research team, during which the advisory group members were asked to weigh in on how we might understand emerging findings. The final contact point was a feedback session during which the research team presented the emerging findings, suggested ways to frame key information and discuss its relevance to them and other youth, and suggested ways to disseminate the findings to youth. An ethics exemption for this review was obtained from Stellenbosch University. However, ethical procedures, including obtaining informed consent to participate from each youth advisory group member, were adhered to throughout the project. Due to their contributions to drafting and reviewing drafts, and particular interest in research, three members of this youth advisory group are represented on the authorship of this paper, Esta Suubi, Shahd Aljassem, and Nawar Aljasem.

The PROSPERO (https://www.crd.york.ac.uk/prospero/ (accessed on 10 December 2022)) record of this collaboratively developed protocol for this review is CRD42021269429 (see Appendix A). Further details on the review methodology are provided below.

### 2.1. Eligibility Criteria

#### 2.1.1. Population

We included intervention studies with before-and-after measures, where adolescents (i.e., aged 14–24 years) constituted at least 50% of the total sample, or where age-disaggregated data could be extracted.

#### 2.1.2. Interventions

We included any intervention targeting adolescents with the aim of improving social inclusion and reducing symptoms of depression and/or anxiety, including community-based interventions and outreach, psychological and counselling support, media campaigns, career development initiatives and educational programmes. Examples of common social inclusion interventions are outlined in recent reviews, and include the following (from Saran et al. [45]):Networking and social support, including linking people to appropriate support networks in the community, for example, non-governmental organisations and self-help groups.Improving community attitudes by working with the media to promote positive images and role models of marginalised groups and making information on services available.Social and communication skill training, including therapeutic approaches used to improve interpersonal relations.Access to, and participation in, cultural programmes, arts, drama and theatres.Access to the legal system and justice.

#### 2.1.3. Control Groups

Where studies were controlled, any control was eligible, including (i) adolescents exposed to other forms of intervention, and usual practice (ii) adolescents not exposed to any intervention. We also included uncontrolled designs, but these needed to include at least two time points.

#### 2.1.4. Types of Studies

We included studies that were designed to assess intervention impact (including, for instance, randomised controlled trials, controlled and uncontrolled before and after designs). Descriptive studies, such as cross-sectional interview studies and single time point surveys, were not included. We included uncontrolled studies, but only if there were two time points (baseline and post-intervention). Only studies published in English were eligible for inclusion.

#### 2.1.5. Setting

Any intervention setting was eligible.

#### 2.1.6. Outcomes

The outcome measures of interest were symptoms of depression and/or anxiety, measured by self-report, clinician-report, or recorded by study investigators. To be included in this review, studies also needed to have an explicit measure of social inclusion, so that we could draw conclusions about the possible role of social inclusion in the intervention’s mechanism of action.

#### 2.1.7. Information Sources

Our review included peer-reviewed, published literature concerning our topic of interest.

#### 2.1.8. Search Strategy

Using a broad search string containing terms related to the population (e.g., adolescen*, child*, teen*), intervention (social inclu*, exclusion, participat*), and outcomes (anx*, depress*) of interest, we searched the following electronic databases:MEDLINE(R);Embase Classic + Embase;PsycINFO;CAB Global Health;CINAHL;ERIC;CENTRAL;Scopus;Web of Science (Social Sciences Citation Index);WHO Global Health Index.

No restrictions in terms of date or format were placed on the search, but only English-language publications were eligible (due to limitations of the review team). We also screened the reference lists of any reviews identified in the search and conducted forward searches of common measures of social inclusion. The full search strategy is available in the Appendix A. It is important to note that our search strategy for this systematic review was very broad. While some reviews of social inclusion interventions have been done before, there have not been many, and, as noted, the construct is multifaceted, and variously defined across past studies. To ensure that all potentially relevant studies were included, we employed a broad search. This search included terms which have been used in the past in the context of social inclusion interventions (such as ‘sense of belonging’ and ‘isolation’), to check whether studies employing these keywords also met our criteria for social inclusion interventions.

#### 2.1.9. Selection Process

We used an online reference management tool, Rayyan (https://www.rayyan.ai/ (accessed on 20 August 2022)) to manage the screening process. All screening was done manually by members of the author group. Screening of each article was done by two reviewers with disagreement resolved by a third reviewer. The full texts of potentially relevant articles were then screened by two reviewers with disagreements resolved by consensus and discussion with the senior authors (XH, JB). Multiple publications of the same study were examined as a single study. The screening process is reported in a PRISMA flow chart below (see Figure 1, below; reasons for exclusion can be obtained from the study authors upon request).

#### 2.1.10. Data Collection Process and Data Items

Two independent reviewers coded the included studies. Data from included studies were extracted using a Microsoft Excel extraction sheet which was piloted on 5 studies prior to use. Coding sheets extracted information related to a range of characteristics of studies and programmes, including study design, intervention type, outcome categories and outcome measures, intervention target, level of delivery, and platform of delivery. The full dataset for this work is available as part of the Appendix A.

#### 2.1.11. Risk of Bias (Confidence in Study Findings) Assessment

To assess risk of bias included studies–otherwise termed confidence in study findings–we used a tool [46] with the following five dimensions: Study design, presence of masking (blinding), attrition, definition of outcome measures, and baseline balance. These domains were defined as follows:Study design (Potential confounders considered): impact evaluations need either a well-designed control group, preferably based on random assignment, or an estimation technique which controls for confounding and the associated possibility of selection bias.Masking (RCTs only, also known as blinding): masking helps limit the biases which can occur if study participants, data collectors or data analysts are aware of the assignment condition of individual participants.Loss to follow up: Attrition can be a major source of bias in studies, especially if these is differential attrition between the treatment and comparison group so that the two may no longer be balanced in pre-intervention characteristics. The US Institute of Education Sciences What Works Clearing House has developed standards for acceptable levels of attrition, in aggregate and the differential, which we applied.Clear definition of outcome measures: this is needed to aid interpretation and reliability of findings and comparability with other studies. Studies should clearly state the outcomes being used with a definition and the basis on which they are measured, preferably with reference to a widely used international standard.Baseline balance shows that the treatment and comparison groups are the same at baseline. Lack of balance can bias the results.

This tool, developed by Howard White from the Campbell Collaboration and Hannah Kuper from London School of Hygiene and Tropical Medicine, with input from Hugh Waddington at 3ie, and previously applied in Campbell Evidence Gap Maps and Systematic Reviews, allowed us to assess the degree to which each study was conducted and reported rigorously, and provided insights about the degree to which those findings could be used to inform the conclusions of this review. Confidence in study findings was rated high, medium, or low, for each of the criteria, applying the standards set by the tool’s creators (see Appendix A). Overall study quality was the lowest rating achieved across the criteria–the weakest link in the chain principle.

#### 2.1.12. Effect Measures

Treatment effect sizes (ES) on depression and anxiety outcomes were calculated as the standardised mean differences (SMDs) between the control and intervention arms with respect to change in unadjusted mean values.

#### 2.1.13. Narrative Synthesis

The extracted data were synthesised narratively. The narrative synthesis described the scope of the literature, as well as the nature of the interventions, the contexts in which they are effective and the target populations for whom they are effective. A narrative summary was prepared for the main themes and findings, including consideration of where there is strong evidence for effect, where there are evidence gaps, and the quality of the evidence.

#### 2.1.14. Forest Plot

Only randomised control trials (RCTs) were included in the generation of forest plots. Treatment effect sizes (ES) on depression and anxiety outcomes were calculated as the standardised mean differences (SMDs) between the control and intervention arms with respect to change in unadjusted mean values at endline. That is, each effect size was calculated by the change in mean differences between the intervention and control groups at the endline divided by their pooled standard deviation. A random-effects model was used, since we assumed that the true treatment effect differs from study to study [47]. Pooled ES estimates were based on a random effects model.

One study [48] contained two treatment arms and one control arm. This study was a cluster RCT and was included after adjustment for design effect. We did, however, calculate the correlation between the two treatment arms and adjust standard errors to reflect this correlation.

Effect sizes of each study were analysed using a forest plot. Greater negative effect sizes represent greater improvements in depression or anxiety symptoms. Heterogeneity of the pooled effect size was assessed using the I^2^ statistic and the Q statistic. In studies that had more than one follow-up assessment, the assessment closest to the end of the intervention was used.

## 3. Results

Our search yielded 117,084 abstracts. The abstracts were deduplicated, and 42,828 duplicates removed. The remaining 74,256 papers were screened by title and abstract by a team of six reviewers, working in pairs. The three pairs screened all papers, with each paper screened by both pair members, independently. Based on this process, 73,825 papers were excluded as their abstract did not indicate that the associated study met inclusion criteria. The remaining 431 manuscripts were then screened on full text. A further 419 studies were excluded because an examination of the full text revealed that the study did not meet inclusion criteria on either population, intervention, design, or location. This resulted in the final set of 12 papers identified for inclusion (Figure 1). Details of included studies are presented in Table 1. All disagreements at both stages of screening were resolved through discussion by XH and JB.

Before proceeding with a discussion of the included studies, it is worth commenting, briefly, on the reason for the large number of excluded papers. Because social inclusion interventions have seldom been subject to systematic review, and because these interventions are extremely diverse in type, we employed a very broad search strategy in this review. However, because little work has been done in this area, we also wanted to ensure that our analyses were rigorous and defensible. As such, we took the decision to ensure that all included studies had both a measure of depression or anxiety and a measure of social inclusion (thereby allowing us to be sure that the active ingredient in which we were interested–social inclusion–was at play in the included studies). This meant that, while we identified a large number of studies on possibly relevant topics, we only included those which we had evidence were leveraging social inclusion as part of their mechanism of action. Due to the combination of a broad search strategy and the rigorous application of inclusion and exclusion criteria, we excluded a large number of studies.

### 3.1. Risk of Bias (Confidence in Study Findings) Results

Risk of bias results are presented in Table 2. Only one study was assessed as being of high quality [6]. Low ratings were largely driven by the use of less rigorous (before versus after) study designs, and poor reporting of masking in RCTs. Only four studies [60,61,62,63] scored medium using our assessment tool, with the remaining eight [64,65,66,67,68,69,70,71] scoring low. Low ratings were largely due to studies employing before versus after designs [64,66,67,68,70,71].

### 3.2. Forest Plot of Intervention Effects

Eight of the 12 included studies were RCTs and thus suitable for inclusion in the forest plot [48,49,50,52,54,56,57,58]. Figure 2 presents forest plots of posttreatment effect sizes, with SMDs and 95% confidence intervals (CIs). Pooled standardised mean differences (SMDs) based on random-effects models showed medium-to-large effects of interventions on improving depression and anxiety symptoms (n = 8; SMD = −0.62; 95% CI, −1.23 to −0.01, *p* < 0.05).

The analysis had considerable heterogeneity (Q = 4359.596.3, *p* = 0.00, I^2^ = 99.71%). Thus, study-level variables were examined narratively to identify potential moderators of the effect on depression and anxiety symptoms.

Since social inclusion was a primary objective, we narratively examined whether improvements in social inclusion outcomes corresponded to greater improvements and depression and anxiety symptoms (see below). Social inclusion outcomes, although heavily varied in their measure, were all significant, with the exception of one study [49]. The strongest effect was evident in a study that included female participants only (see Gee et al. [52]).

### 3.3. Intervention Modalities and Mechanisms of Change

Social skills training in small groups was the most common modality used to promote social inclusion. Social skills training was typically integrated with other modalities, including mindfulness and yoga breathing exercises [50]; cognitive and interpersonal therapy [58]; systemic family therapy [51], and a holistic psychoeducation programme [53]. Social skills training was also integrated into expressive arts-based interventions, such as song writing workshops [52] and hip-hop groups [55]. The evidence we have reviewed indicates that social skills training may be an important component of social inclusion interventions to promote adolescent mental health, however, it is not clear whether teaching social skills on its own would be a sufficient condition for promoting social inclusion. It is possible that the positive effects on social inclusion observed were, at least in part, a consequence of the group setting in which the interventions were delivered. While social skills training could plausibly be an effective means of promoting social inclusion, for instance among special populations such as adolescents on the autistic spectrum (who typically have difficulties interpreting social cues) [59], it is unclear how effective this might be for adolescents who face concrete structural barriers to participation in civic life. For instance, where adolescents are living in poverty, or facing systematic discrimination on the basis of identity (i.e., racism, homophobia), it seems unreasonable to believe that improved social skills on their part could lead to improved social inclusion, as the drivers of exclusion lie outside of the individual.

Life skills programmes delivered in schools as part of the curriculum were shown to be effective for promoting social inclusion in two studies [48,54]. These psychoeducational life skills programmes included a range of different topics, such as communication skills, information about the process of growing up, relationship building skills, gender and sexuality education, and information about substance use. From the limited available evidence, it seems that life skills interventions to promote social inclusion could be effectively delivered by teachers and lay-counsellors and need not necessarily be delivered by trained mental health professionals, which could have important implications for implementation in low-resource settings. It is, however, unclear what the optimal duration and dosage is for these interventions to be effective and what content needs to be included to promote social inclusion. It is also not yet apparent if life skills programmes need to be implemented school-wide for them to be effective.

Cash transfers were used in one intervention [49]. This study showed that an unconditional cash transfer programme targeting ultra-poor adolescents can promote social inclusion and reduce depression and anxiety. It seems highly probable that cash transfers would promote social inclusion by providing the means for individuals to participate more fully in economic life. It is not immediately clear, however, how improving adolescents’ participation in the economy via cash transfers would result in increases in perceived social support, as was demonstrated by Angeles et al. [49]. It is also not clear from the limited evidence available to determine the minimum threshold (i.e., the minimum amount of money) needed for a cash transfer to be effective at improving social inclusion sufficiently to show reductions in depression and anxiety.

Only one intervention explored a mentorship programme, within a positive youth development framework, to promote social inclusion and reduce adolescent depression and anxiety [56]. Mentorship programmes which connect vulnerable youth to other young people with greater social capital maybe an effective way to promote inclusion by providing isolated adolescents with a bridge into social spaces and social networks which they would not otherwise be able to access easily. It seems probable that mentorship interventions may be particularly effective for adolescents in transition (e.g., those entering new schools) and adolescents who do not have access to social capital (e.g., refugees arriving in new countries), although we do not have the evidence to know if this would be the case. Furthermore, it is not yet clear from the available evidence what the optimal duration is for a mentorship programme to be effective, what skills and training mentors require to be effective, and what mentor characteristics (e.g., social standing/status) are required to optimise the effectiveness of these kinds of interventions.

Youth empowerment was an explicit goal in only one intervention [55], despite the obvious potential to promote social inclusion of marginalised adolescents by increasing their agency, authority, and autonomy. Interestingly, the empowerment intervention we identified engaged adolescents via a hip-hop beat-making programme, which may, in part, have been effective as a social inclusion intervention because it provided adolescents with direct access to a means of participating in a youth cultural activity (i.e., social capital). There may be other mechanisms to empower adolescents, for example through enabling participation in political activities as is done in the model United Nations project (www.un.org/en/mun (accessed on 10 December 2022)). More research is needed to explore the effectiveness of empowerment interventions to prevent and treat adolescent anxiety and depression, particularly to understand for whom these kinds of interventions are most appropriate.

Finally, one intervention made use of reading–writing exercises to promote social inclusion by normalising adolescents’ experience of struggling to find belonging [57]. Bibliotherapy and therapeutic storytelling may be an effective way to promote social inclusion provided the content of these materials is focused on social dynamics which create and perpetuate marginalisation. One potential advantage of bibliotherapies is that they can be easily integrated into the school curriculum and can be delivered by teachers, although it remains to be seen if these interventions are effective at increasing social inclusion, whether they are acceptable to adolescents, and whether they are feasible on a large scale (especially in resource-constrained environments). Critically, we will need to understand what content and themes are needed for texts to be used therapeutically to counter social exclusion.

It is clear from the studies we identified that a range of modalities have been successfully used to promote adolescent mental health and treat mental health problems by increasing social inclusion. However, the mechanisms by which these interventions achieve their outcomes are poorly understood. Furthermore, the necessary and sufficient conditions required to increase social inclusion and reduce depression and/or anxiety are unclear.

## 4. Discussion

While there is convincing data to show that socially excluded adolescents are at increased risk of depression and anxiety, there is only a modest body of research to support the use of interventions promoting social inclusion to prevent and treat these common mental health problems. Despite extensive and systematic literature searches, we were only able to identify 12 studies which investigated the effectiveness of interventions that promote social inclusion and reduce symptoms of depression and/or anxiety among adolescents. This is partly because many interventions claiming to act upon social inclusion, do not actually measure it, and so were excluded from this review. Nonetheless, the available evidence suggests that promoting social inclusion could be an effective means of addressing high rates of anxiety and depression among adolescents globally.

The interventions we identified primarily focused on equipping adolescents to navigate their social environments by enhancing their social skills. As such, we do not have much evidence on how to prevent or treat adolescent depression and/or anxiety by modifying physical and social environments to promote social inclusion. The bias towards focusing mental health interventions on individuals rather than on eco-systems or environments reflects psychiatry’s long history of assuming that psychopathology originates within individuals rather than in the spaces between them.

As might be expected in a review focused on social inclusion, several of the studies we identified described interventions targeted at young people at risk of exclusion, such as those with disabilities or experiencing poverty. In the same vein, the literature included in our review is weighted in its focus on the subjective inter-personal dimensions of social inclusion, such as connection, belonging, and quality of social interactions, but less so on the socio-political dimensions of social inclusion, such as access to justice and improved economic participation, which are well-documented but under-researched in relation to youth. As noted, studies were only eligible for inclusion if they included a social inclusion intervention, had a measure of social inclusion, and measured depression and/or anxiety as an outcome. As such, the absence of studies addressing these factors does not reflect a general absence of effort in this area; indeed, programmes addressing the social determinants of mental health among youth exist. However, based on the findings of this review, such programmes are not measuring social inclusion or mental health systematically. Conversely, programmes may well be measuring mental health among youth and binary social inclusion outcomes (such as ‘enrolled in school’, ‘gained a job’, ‘voted’) but if the associated intervention is not a social inclusion intervention (see Saran et al. [45]), then it is difficult to argue that the social inclusion outcomes are due to efforts to improve social inclusion, and not some other factor. It is of course not easy to tease out the mechanisms through which psychosocial interventions are effective but attempting to do so could provide valuable insights into the role of social inclusion as an effective ingredient in interventions for adolescents.

The populations targeted in the interventions we identified included economically marginalised adolescents, youth with peer social problems, and youth with autism spectrum disorders. We did not identify any studies which examined depression/anxiety and social inclusion interventions explicitly targeted at refugee or LGBTQ+ youth, despite significant amounts of evidence to suggest that these adolescents are at risk of social exclusion. It is notable that the largest effect of an intervention was seen in a study which included only female participants, raising the question of possible gender effects of these kinds of interventions. Given the known vulnerability of female adolescents in terms of risk for psychopathology and sensitivity to social rejection, it is possible that social inclusion interventions have a differentially positive effect for this group.

It is concerning that many studies we identified utilised exclusion criteria which systematically excluded vulnerable young people from participating in the interventions. Exclusion on certain grounds may be methodologically warranted in some cases (for instance, insisting on recruiting only youth who are not currently enrolled in another study). However, it was not always clear whether exclusion for certain reasons (e.g., severe mental health problems) was warranted in all the studies which employed these criteria. The net effect of excluding adolescents on the basis of certain risk factors, in many of the studies, is that there are particular groups of adolescents (such as those with intellectual or psychosocial disabilities) who, despite being at increased risk for social exclusion, have been systematically omitted from to improve social inclusion.

The limitations of the studies we identified are notable and indicate areas where the scientific rigour in intervention research focused on adolescent mental health and social inclusion could be improved. It is imperative that future studies in this area are well designed to avoid a risk of bias. Large, well-designed intervention studies are of course expensive and time consuming, nonetheless the results of this review strongly suggest that these trials are warranted given the promising effects we found.

## 5. Conclusions

The body of evidence to support social inclusion as a key ingredient of interventions to prevent or treat adolescent depression and/or anxiety is too small to draw firm conclusions. However, the work that has been done in this area is promising and indicates that ongoing research is warranted. Rigorous studies with less restrictive exclusion criteria and reliable and accurate measures of social inclusion are needed to deepen our understanding of the links between adolescent mental health and social exclusion. The development of theories to explain the mechanisms of change and elucidate how promoting social inclusion leads to reductions in depression and anxiety is integral to advancing science and practice in this area of adolescent mental health.

### Limitations

The systematic literature searches, the clear definition of core concepts, the careful extraction and analysis of data, and enlisting the participation of a youth advisory group are significant strengths of this review. Nonetheless, it is a limitation that we only included peer-reviewed studies published in English. Therefore, because of the large number of abstracts and the large number of reviewers, as well as limitations of the software used for screening, no tests of inter-rater reliability were conducted. While there were not a great number of disagreements, and all disagreements were carefully and thoughtfully resolved, this is a limitation. Additionally, by omitting the grey literature and literature published in other languages, we may indeed have missed eligible works. For the forest plot, limitations include large heterogeneity across interventions and a lack of standardised measures for depression and anxiety. The advisory group members also represent a narrow age band (21–24 years) within the broad age range (14–24 years) with which our study was concerned. This was partly due to the relative ease, from a research ethics perspective, of obtaining consent to work with older adolescents, and because the advisory group was drawn from the knowledge networks of the first author, who mostly works with older youth. The lack of representation of younger adolescents, and adolescents from the same countries as the identified studies, in the advisory group may have affected the types of feedback they provided during the research process. Finally, we included only studies which had an eligible measure of mental health and an eligible measure of social inclusion. Many studies deliver interventions which aim to shift social inclusion or mental health but do not measure social inclusion or mental health as outcomes. We chose not to include these studies because deeming them eligible without evidence or measurement of seemed hard to defend conceptually. However, narrowing the focus of the review to studies with measured outcomes in both domains excludes some of the literature which may be relevant for fully understanding these types of programmes and their impact.

## Figures and Tables

**Figure 1 ijerph-20-01895-f001:**
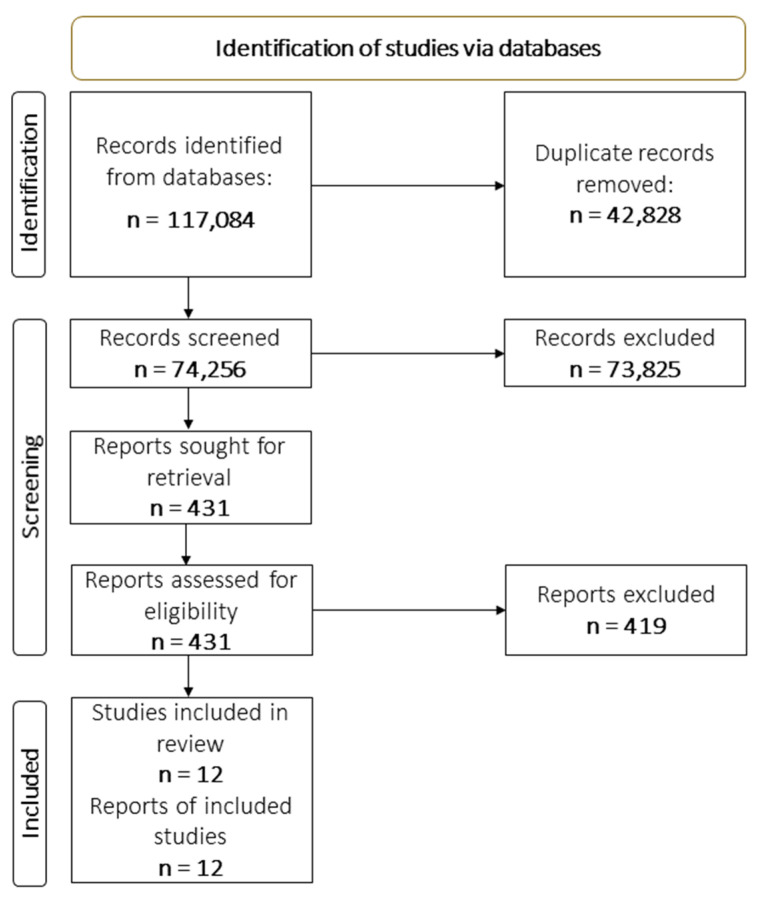
PRISMA Flowchart. Detailed reasons for exclusions of studies are available from the authors upon request.

**Figure 2 ijerph-20-01895-f002:**
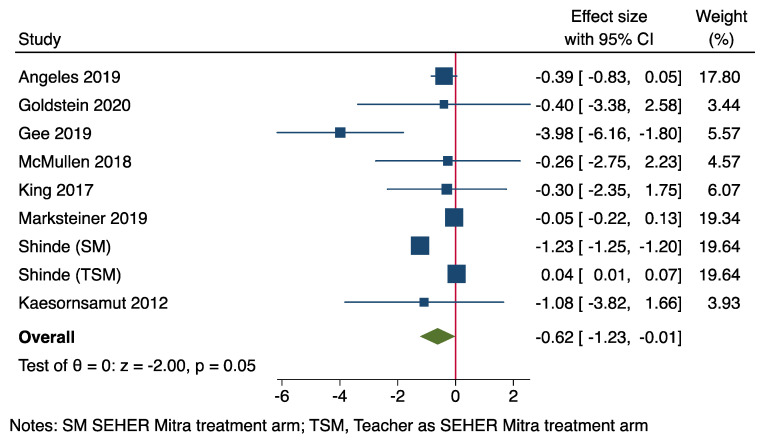
Forest plot for effect of social inclusion interventions on adolescent depression and anxiety symptoms [48,49,50,52,54,56,57,58].

**Table 1 ijerph-20-01895-t001:** Summary of included studies.

		Angeles 2019 [49]	Goldstein 2020 [50]	Curtis 2018 [51]	Gee 2019 [52]	Hughes 2021 [53]	McMullen 2018 [54]	Travis 2019 [55]	King 2017 [56]	Marksteiner 2019 [57]	Shinde 2020 [48]	Kaesornsamut 2012 [58]	Hill 2017 [59]
	Sample size	2099	69	107	13	39	170	35	218	106	7824	60	5
**Level**	Universal		x				x	x		x	x		
	Targeted	x							x				x
	Indicated			x	x	x						x	
	Age Range	13–19 years	18–35 years	12–17 years	18–31 years	18–25 years	13–18 years	11–15 years	12–15 years	19.8 years *	13–15 years	16–18 years	13–15 years
**Gender**	Female Only				x								
	Male and Female	x	x	x		x	x	x	x	x	x	x	x
	Country	Malawi	USA	USA	UK	USA	Uganda	Australia	USA	Germany	India	Thailand	USA
**Outcome**	Depression		x	x	x	x	x	x	x	x	x	x	x
	Anxiety		x	x	x	x	x	x					x

* No age range given; mean reported. Universal interventions are those which are delivered to all individuals in a given population, for instance, a whole class or school or town. Targeted interventions are those where participants are targeted based on some pre-existing risk factor which suggests that they are at greater need for intervention. Indicated interventions are for individuals who are already experiencing the outcome targeted by the intervention, for example, depression or anxiety.

**Table 2 ijerph-20-01895-t002:** Risk of bias results.

		Angeles 2019 [49]	Goldstein 2020 [50]	Curtis 2018 [51]	Gee 2019 [52]	Hughes 2021 [53]	McMullen 2018 [54]	Travis 2019 [55]	King 2017 [56]	Marksteiner 2019 [57]	Shinde 2020 [48]	Kaesornsamut 2012 [58]	Hill 2017 [59]
**Study Design**	Low			x		x		x					x
	Medium						x						
	High	x	x		x				x	x	x	x	
**Loss to Follow Up**	Low	x	x				x		x				
	Medium			x	x	x				x			
	High							x			x	x	x
**Masking**	Low	x	x	N/A	x	N/A	N/A	N/A	x	x		N/A	N/A
	Medium												
	High										x		
**MH Outcome Measure**	Low												
	Medium			x									
	High	x			x	x	x	x	x	x	x	x	x
**Baseline Balance**	Low			N/A	x			x					N/A
	Medium	x				x	x						
	High		x						x	x	x	x	
**Overall score**	Low	x	x	x	x	x	x	x	x	x		x	x
	Medium												
	High										x		

## Data Availability

The data presented in this study are available in Appendix A.

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
