# Peer review of "Effectiveness of Social Inclusion Interventions for Anxiety and Depression among Adolescents: A Systematic Review"

_ijerph, 2023, doi:10.3390/ijerph20031895_

Round 1

Reviewer 1 Report

Excently written and interesting read, with just 4 minor comments / revisions:

1. I would encourage changing the title as I would not describe this as participatory research methods / participatory systematic review - it is a systematic review with lived experience involvement of young people via online consultation, unless they did conduct some of the systematic review processes, in which case, I would encourage emphasising this!

2. I am unsure of the use of Rayyan for title/abstract screening, which is typically done manually, particulary with the lack of a reference for it's validation / reliability. Please provide more detail / reference / prior examples of it's use. 

3. Whilst the authors indicate how disagreements were resolved, there does not appear to be any indication that inter-rater reliability testing was conducted?

4. Please provide more details regarding the quality tool - from the supplementary materials, I'm still unsure as to who the the tool's creator is, and again whether it is reliable or validated - it is typical to use SAQOR or GRADE in systematic reviews. 

Author Response

Thank you very much for your positive feedback on this paper and for your thoughtful comments. Please find below our responses to your requested revisions.

  1. I would encourage changing the title as I would not describe this as participatory research methods / participatory systematic review - it is a systematic review with lived experience involvement of young people via online consultation, unless they did conduct some of the systematic review processes, in which case, I would encourage emphasizing this!

Thank you for raising this. We have amended the title, and nuanced the explanation of their role/the lived experience component (removing reference to participatory methods and calling it lived experience involvement instead).

  1. I am unsure of the use of Rayyan for title/abstract screening, which is typically done manually, particularly with the lack of a reference for it's validation / reliability. Please provide more detail / reference / prior examples of it's use.

Sorry for not clarifying – everything was done manually. Rayyan is simply an online management software, like EppiReviewer or Covidence. The website address (Rayyan.ai) is misleading as it does not have any AI capabilities. We have clarified this in the text.

  1. Whilst the authors indicate how disagreements were resolved, there does not appear to be any indication that inter-rater reliability testing was conducted?

This is certainly a limitation of the study. Unfortunately, because the screening team comprised >6 individuals, there were a very large number of abstracts, the review software does not have in-built inter-rater reliability functionality, and the timelines for the review (which was commissioned by Wellcome Trust) were very tight, we did not calculate inter-rater reliability. We forgot to include this in the limitations section, and apologize for the omission, as it is an important thing to note. We have included it there, now.

  1. Please provide more details regarding the quality tool - from the supplementary materials, I'm still unsure as to who the the tool's creator is, and again whether it is reliable or validated - it is typical to use SAQOR or GRADE in systematic reviews.

We have now included much more detail on the tool, including its creators and prior applications.

Reviewer 2 Report

This paper reports on a participatory systematic review of the effectiveness of social inclusion interventions for anxiety and depression among adolescents.  Authors assert that adolescents who are socially excluded from society due to such acts as discrimination, prejudice, disempowerment, and stigmatization are at increased of mental health concerns like depression and anxiety and psycho-social issues like poverty and lack of educational and employment opportunities.  Authors also contend that interventions that promote social inclusion can help prevent and treat adolescent depression and anxiety and promote adolescent mental health.  Therefore, authors conduct a systematic review of intervention studies that aim to prevent or treat adolescent depression and/or anxiety and promote social inclusion and report on their findings.  The review is as follows:

1)     Line 37 – Explain the term ‘intra-psychic’ for the lay reader

2)     Line 46 – Explain what is meant by the term ‘political purchase’.

3)     Lines 65-66 – In “The term “social inclusion” is also used to describe the consequences of increasing people’s access to society…”, might there be a better word to choose instead of ‘consequences’, which may have a negative connotation?

4)     In the Materials and Methods, the engagement of a youth advisory group of 13 young people from countries in Africa and the Middles East and their inclusion in the paper’s authorship is commendable.

5)     Lines 306-310 – In “While social skills training could plausibly be an effective means of promoting social inclusion, particularly among special populations like adolescents on the autistic spectrum (who typically have difficulties interpreting social cues) [71], it is unclear how effective this might be for adolescents who face concrete structural barriers to participation in civic life”, give examples of what would be ‘concrete structural barriers to participation in civic life’.

6)     Lines 388-391 – There is a good mention of the need for mental health interventions to focus on environment and ecosystems rather than on individuals.

7)     Line 418-422 – There is an intriguing and insightful discussion of the possible gender effects of social inclusion intervention for adolescents and the vulnerability of female adolescents and sensitivity to social rejection.

8)     Lines 433-435 – In “The poor quality of the studies we identified is notable and indicates a lack of scientific rigour in intervention research focused on adolescent mental health and social inclusion”, the use of statements ‘poor quality’ and lack of scientific rigor’ seems a bit disparaging.  Authors should consider using alternative words that are less harsh (e.g., partial, or limited quality; partial or limited rigor).

Overall, this is an insightful, unique study on a pertinent topic.  The paper is comprehensive and well-research.  The methodology is sound.  Addressing some clarifying questions may help the paper to become clearer and improved.

Author Response

Thank you very much for your feedback. It is a great experience, as authors, to read such concrete and constructive feedback on a paper. We appreciate the time and effort you took to review this piece and believe that the changes you have suggested will strengthen the paper.

1)        Line 37 – Explain the term ‘intra-psychic’ for the lay reader

Thank you. We have included the following definition “psychological phenomena that arise or occur within the psyche or mind”.

2)        Line 46 – Explain what is meant by the term ‘political purchase’.

We have simplified the sentence to work without the use of this phrase.

3)        Lines 65-66 – In “The term “social inclusion” is also used to describe the consequences of increasing people’s access to society…”, might there be a better word to choose instead of ‘consequences’, which may have a negative connotation?

We have rephrased the sentence to “The term “social inclusion” is also used to describe what happens when people’s access to society increases”.

4)        In the Materials and Methods, the engagement of a youth advisory group of 13 young people from countries in Africa and the Middles East and their inclusion in the paper’s authorship is commendable.

Thank you for highlighting this. We were very lucky to have them join this work.

5)        Lines 306-310 – In “While social skills training could plausibly be an effective means of promoting social inclusion, particularly among special populations like adolescents on the autistic spectrum (who typically have difficulties interpreting social cues) [71], it is unclear how effective this might be for adolescents who face concrete structural barriers to participation in civic life”, give examples of what would be ‘concrete structural barriers to participation in civic life’.

Thank you for requesting this clarification. We have now given the following example: “For instance, where adolescents are living in poverty, or facing systematic discrimination on the basis of identity, it seems unreasonable to believe that improved social skills on their part could lead to improved social inclusion, as the drivers of exclusion lie outside of the individual.”

6)        Lines 388-391 – There is a good mention of the need for mental health interventions to focus on environment and ecosystems rather than on individuals.

Thank you for the positive feedback. The data seem to support that it is important for these kinds of intervention receive greater attention.

7)        Line 418-422 – There is an intriguing and insightful discussion of the possible gender effects of social inclusion intervention for adolescents and the vulnerability of female adolescents and sensitivity to social rejection.

Thank you again for highlighting sections of the paper which you found stimulating. This is very encouraging.

8)        Lines 433-435 – In “The poor quality of the studies we identified is notable and indicates a lack of scientific rigour in intervention research focused on adolescent mental health and social inclusion”, the use of statements ‘poor quality’ and lack of scientific rigor’ seems a bit disparaging.  Authors should consider using alternative words that are less harsh (e.g., partial, or limited quality; partial or limited rigor).

We have reworded this section to use less disparaging terms. Thank you for drawing attention to this.

Overall, this is an insightful, unique study on a pertinent topic.  The paper is comprehensive and well-research.  The methodology is sound.  Addressing some clarifying questions may help the paper to become clearer and improved.

Thank you!

Reviewer 3 Report

The manuscript reports the findings of a systematic review of the efficacy of interventions targeting social inclusion on depression and anxiety among adolescents. This is a novel and potentially insightful research question, shedding new light on the understanding of community-based and psychosocial interventions for depression and anxiety in adolescents. I have two main concerns about this manuscript, as follows:

 -        The definition and operationalization of social inclusion are of critical importance to this review. In the Introduction, the authors offered various perspectives and/or definitions of social inclusion (and exclusion), and mentioned that “However, the term is contested and has been variously used to denote a variety of related constructs” (lines 47-48, p. 2). It would be important to clarify what these “related constructs” are, for the readers’ understanding of what social inclusion is and is not. Relatedly, I saw that numerous search terms related to social inclusion were used (Supplementary material 1), including social acceptance, sense of belonging, social capital, social isolation, social network, and even loneliness. I think these are related but distinct concepts. That had made the search over-inclusive (this was not made clear in the pre-registration protocol as well). This point is highly critical, as the operationalization of social inclusion defines the inclusion criteria of the review and, thus, what studies would enter the pool of studies for subsequent qualitative and quantitative syntheses. In particular, in the Discussion section 3.5, the authors mentioned “Social skills training in small groups was the most common modality used to promote social inclusion” (lines 296 -297, p.9). I am not sure how social skills training is directly related to an improvement in social inclusion. A strong justification is needed for such an over-inclusive search.

 -        The value and/or importance of the “participatory” component of the review, which is the strength of this review, could be further elaborated and extended in the whole manuscript (e.g. in the last paragraph of the Introduction (lines 110-119, line 3). Maybe some relevant points in the Method could be moved to the Introduction as an overview, informing the readers of the significance of such an approach earlier in the manuscript. I appreciate that a youth advisory group “was engaged by the research team during key steps in the review process which required youth input to ensure the review and its findings include the reflections and expertise of young people.” (lines 125-127, p. 3). I would like to know the exact involvement and contribution of the advisoy group to each stage of the review process to make this methodology meaningful and the findings valid. Therefore, it would be relevant to know how their comments and feedback were incorporated as an integral part of the review process. The limitation (session 5.1, lines 450- 471, p. 12) mentioned a few of their involvement, and it is not very clear about the contribution of the group to the current findings of the review.

 Some additional comments for specific sections of the manuscript are as follows:

 Introduction:

-        About the statement “While there is some general agreement in the literature on the processes by which people may become socially excluded, less attention has been paid to identifying mechanisms to promote social inclusion [38] (although some recent exceptions exist, see Gardner, et al. [40]).” (lines 92-95, p. 2), further elaboration of these mechanisms would be informative for the readers to appreciate the efficacy of various types of interventions in increasing social inclusion (or reducing social exclusion). The examples in the “Intervention(s), exposure(s)” section of the PROSPERO protocol would also be helpful to illustrate this point. Also, a few examples of these interventions, perhaps some more specific ones under section 2.1.2, would make the inclusion criteria more self-explanatory and concrete.

 Method:

-        For section 2.1.14, I am curious about the rationale for the inclusion of only randomized trials in the meta-analysis. So the non-randomized trials were just for the qualitative synthesis?

-        An analysis of the publication bias, such as a funnel plot, should also be included in a meta-analysis.

 Results:

-        Table 1 should also include more details of the included studies, such as social inclusion interventions of the included studies, measures of social inclusion, and measures of depression and anxiety. These are relevant study characteristics extracted from the included studies for an overview of how interventions targeting social inclusion could improve depression and anxiety, ass the core research question of this review. Also, the meaning of “universal”, “targeted”, and “indicated” for the row of “level” is not obvious, so some explanation of these terms would be useful.

-        How can we make sense of the high statistical heterogeneity of the pooled standardized mean difference? Could this be due to a broad operationalization of social inclusion and various type of interventions, as well as the inclusion of both general and specific populations (i.e. youth with disabilities)?

 Supplementary materials:

I found that S1 and S2 are the same files. Also, I couldn’t open S3. I suggest the authors double-check these files again.

Author Response

The manuscript reports the findings of a systematic review of the efficacy of interventions targeting social inclusion on depression and anxiety among adolescents. This is a novel and potentially insightful research question, shedding new light on the understanding of community-based and psychosocial interventions for depression and anxiety in adolescents.

Thank you so much for your feedback on the paper. We are very grateful for your input, and hope that the revisions we have made, and the additional information we have provided, are sufficient to address your concerns.

I have two main concerns about this manuscript, as follows:

 -          The definition and operationalization of social inclusion are of critical importance to this review. In the Introduction, the authors offered various perspectives and/or definitions of social inclusion (and exclusion), and mentioned that “However, the term is contested and has been variously used to denote a variety of related constructs” (lines 47-48, p. 2). It would be important to clarify what these “related constructs” are, for the readers’ understanding of what social inclusion is and is not. Relatedly, I saw that numerous search terms related to social inclusion were used (Supplementary material 1), including social acceptance, sense of belonging, social capital, social isolation, social network, and even loneliness. I think these are related but distinct concepts. That had made the search over-inclusive (this was not made clear in the pre-registration protocol as well). This point is highly critical, as the operationalization of social inclusion defines the inclusion criteria of the review and, thus, what studies would enter the pool of studies for subsequent qualitative and quantitative syntheses. In particular, in the Discussion section 3.5, the authors mentioned “Social skills training in small groups was the most common modality used to promote social inclusion” (lines 296 -297, p.9). I am not sure how social skills training is directly related to an improvement in social inclusion. A strong justification is needed for such an over-inclusive search.

Thank you for raising this. We have included two large sections, as well as detail to the introduction section on related constructs, to explain why decisions were made concerning search strategy, and then inclusion and exclusion criteria. Please see the sections below:

Starting line 193:

It is important to note that our search strategy for this systematic review was very broad. While some reviews of social inclusion interventions have been done before, there have not been many, and, as noted, the construct is multifaceted, and variously defined across past studies. To ensure that all potentially relevant studies were included, we employed a broad search. This search included terms which have been used in the past in the context of social inclusion interventions (such as ‘sense of belonging’ and ‘isolation’), because we wanted to be able to check whether studies employing these keywords also met our criteria for social inclusion interventions. The use of these terms in the search string does not imply that we think of these constructs as synonymous, but that we were cautious not to omit any potentially relevant work.

Starting line 297:

Before proceeding with a discussion of the included studies, it is worth commenting, briefly, on the reason for the large number of excluded papers. Because social inclusion interventions have seldom been subject to systematic review, and because these interventions are extremely diverse in type, we employed a very broad search strategy in this review. However, also because little work has been done in this area, we wanted to ensure that our analyses were rigorous and defensible. As such, we took the decision to ensure that all included studies had both a measure of depression or anxiety and a measure of social inclusion (thereby allowing us to be sure that the active ingredient in which we were interested – social inclusion – was in fact at play in the included studies). This meant that, while we identified a large number of studies on possibly relevant topics, we only included those which we were sure were leveraging social inclusion as part of their mechanism of action. Due to the combination of a broad search strategy and the rigorous application of inclusion and exclusion criteria, we excluded a large number of studies. 

Finally, regarding the social skills training. Past systematic reviews and protocols for systematic reviews on social inclusion have included social skills training interventions, please see, for instance:

  1. Louw, J. S., Kirkpatrick, B., & Leader, G. (2020). Enhancing social inclusion of young adults with intellectual disabilities: A systematic review of original empirical studies. Journal of Applied Research in Intellectual Disabilities, 33(5), 793-807.
  2. Saran, A., White, H., & Kuper, H. (2020). Evidence and gap map of studies assessing the effectiveness of interventions for people with disabilities in low‐and middle‐income countries. Campbell Systematic Reviews, 16(1), e1070.
  3. Saran, A., Hunt, X., White, H., & Kuper, H. (2021). PROTOCOL: Effectiveness of interventions for improving social inclusion outcomes for people with disabilities in low‐and middle‐income countries: A systematic review. Campbell Systematic Reviews, 17(3), e1191.

So, there is precedent for thinking of these types of programs as contributing to social inclusion by improving social excluded individual’s ability to navigate complex or exclusionary social environments. See, for instance:

  1. Iarocci, G., Yager, J., Rombough, A., & McLaughlin, J. (2008). The development of social competence among persons with Down syndrome: From survival to social inclusion. International review of research in mental retardation, 35, 87-119.
  2. Mikami, A. Y., Jia, M., & Na, J. J. (2014). Social skills training. Child and Adolescent Psychiatric Clinics, 23(4), 775-788.

. Now, to our understanding, this is not unproblematic, as these types of programs place the onus of social inclusion on the marginalised group. We have mentioned this in the paper (see the section “While social skills training could plausibly be an effective means of promoting social inclusion, particularly among special populations like adolescents on the autistic spectrum (who typically have difficulties interpreting social cues) [71], it is unclear how effective this might be for adolescents who face concrete structural barriers to participation in civic life. For instance, where adolescents are living in poverty, or facing systematic discrimination on the basis of identity, it seems unreasonable to believe that improved social skills on their part could lead to improved social inclusion, as the drivers of exclusion lie outside of the individual.”). Yet, in this review, we included such interventions if they included relevant measures of social inclusion and of mental health. We think this is important because it allows the review to highlight that interventionists are too infrequently focusing on the environmental and systematic drivers of exclusion, and instead focusing on ‘equipping the marginalized to be included’, which is too individualized an approach.

 -          The value and/or importance of the “participatory” component of the review, which is the strength of this review, could be further elaborated and extended in the whole manuscript (e.g. in the last paragraph of the Introduction (lines 110-119, line 3). Maybe some relevant points in the Method could be moved to the Introduction as an overview, informing the readers of the significance of such an approach earlier in the manuscript. I appreciate that a youth advisory group “was engaged by the research team during key steps in the review process which required youth input to ensure the review and its findings include the reflections and expertise of young people.” (lines 125-127, p. 3). I would like to know the exact involvement and contribution of the advisory group to each stage of the review process to make this methodology meaningful and the findings valid. Therefore, it would be relevant to know how their comments and feedback were incorporated as an integral part of the review process. The limitation (session 5.1, lines 450- 471, p. 12) mentioned a few of their involvement, and it is not very clear about the contribution of the group to the current findings of the review.

Thank you for raising this. We had feedback from another reviewer who suggested that we not call this a participatory systematic review, but rather be more modest and call it a systematic review with lived experience involvement. We have gone with this strategy and removed the word participatory from the title. However, we strongly believe that the lived experience involvement on the advisory group, including through their authorship of this paper, was essential to our successful completion of this work, and so we have added detail on these practices in the methods. We have also moved some sections on the importance of lived experience involvement higher up in the manuscript as you suggest.

 Some additional comments for specific sections of the manuscript are as follows:

 Introduction:

-           About the statement “While there is some general agreement in the literature on the processes by which people may become socially excluded, less attention has been paid to identifying mechanisms to promote social inclusion [38] (although some recent exceptions exist, see Gardner, et al. [40]).” (lines 92-95, p. 2), further elaboration of these mechanisms would be informative for the readers to appreciate the efficacy of various types of interventions in increasing social inclusion (or reducing social exclusion). The examples in the “Intervention(s), exposure(s)” section of the PROSPERO protocol would also be helpful to illustrate this point. Also, a few examples of these interventions, perhaps some more specific ones under section 2.1.2, would make the inclusion criteria more self-explanatory and concrete.

This is a very helpful comment, thank you. We have added to the section on interventions

 Method:

-           For section 2.1.14, I am curious about the rationale for the inclusion of only randomized trials in the meta-analysis. So the non-randomized trials were just for the qualitative synthesis?

Our methodologist (GJ Melendez-Torres) is of the school of thought that only RCTs should be included in meta-analyses. This has been standard practice in statistics (see https://systematicreviewsjournal.biomedcentral.com/articles/10.1186/s13643-015-0133-0) for some time, although it is changing. While it is possible to meta-analyse some non-randomised studies, those non-randomised studies included in this review had methodological limitations which our team worried would limit the credibility of the meta-analysis.

Results:

-           Table 1 should also include more details of the included studies, such as social inclusion interventions of the included studies, measures of social inclusion, and measures of depression and anxiety. These are relevant study characteristics extracted from the included studies for an overview of how interventions targeting social inclusion could improve depression and anxiety, ass the core research question of this review. Also, the meaning of “universal”, “targeted”, and “indicated” for the row of “level” is not obvious, so some explanation of these terms would be useful.

We have included definitions of these terms in the legend for the table. However, because we discuss all of the interventions in depth in the narrative synthesis, we worry that it is a duplication (and will push us well over the journal’s allowable word count) to reproduce this information here. If you feel strongly that it belongs in the table, we can change it, but would need to editor’s approval to go above the word count.

-           How can we make sense of the high statistical heterogeneity of the pooled standardized mean difference? Could this be due to a broad operationalization of social inclusion and various type of interventions, as well as the inclusion of both general and specific populations (i.e. youth with disabilities)?

We think that is likely the case. If you would like, we can put a footnote in to gesture to this.

Supplementary materials:

I found that S1 and S2 are the same files. Also, I couldn’t open S3. I suggest the authors double-check these files again.

We changed S 1 and S 2 and ensured that S 3 is a PDF and so should be possible to open using any PDF reader.

Reviewer 4 Report

The intention of this study is relevant and practically significant. The authors have done a large amount of work on the search and analysis of relevant literature. However, in my opinion, the research procedure contains significant shortcomings that do not meet the requirements for systematic reviews:

1)      The authors do not provide in the text of the article the search query with which 117,084 abstracts were selected

2)      The authors need to clarify the purpose of the study: is it a systematic review of the literature or a meta-analysis of studies. After answering this question, it will be necessary to clarify the title of the article and the research procedure

3)      The authors need to justify the eligibility criteria, perhaps this will allow more articles to be included in the review.

The most important question that raises doubts and requires detailed justification: why were so many articles excluded? Is it possible that the original search query was formulated incorrectly or that the inclusion criteria were too strict?

Author Response

The intention of this study is relevant and practically significant. The authors have done a large amount of work on the search and analysis of relevant literature. However, in my opinion, the research procedure contains significant shortcomings that do not meet the requirements for systematic reviews.

We appreciate your feedback and constructive criticism on the piece. Below, we detail our responses to your concerns, many of which we feel we are able to address. We would greatly appreciate if you would reconsider the piece for publication, in light of our responses and the additional information we have provided.

1)     The authors do not provide in the text of the article the search query with which 117,084 abstracts were selected

In line with the journal’s requirements, we have put this information in the supplementary materials, which you should have had access to. We apologise if they were not part of the review pack which you were sent. We would be happy to include them in the main text, or as a footnote, but will need to take this up with the editor due to word count limitations. Nonetheless, the search strategy was provided.

2)     The authors need to clarify the purpose of the study: is it a systematic review of the literature or a meta-analysis of studies. After answering this question, it will be necessary to clarify the title of the article and the research procedure

We are not sure that we understand this comment. All meta-analyses are, by definition, based on a systematic review process. Sometimes, however, systematic reviews do not include meta-analyses. We have done a systematic review and included an analysis of effect sizes. So, we are calling the paper a systematic review, and not claiming that it is a meta-analysis. In terms of clarifying the purpose of the study, we refer the reviewer to the following section,

“We undertook a systematic review of evidence exploring the effectiveness of in-terventions to prevent or treat adolescent depression and/or anxiety by promoting social inclusion, with a focus on answering the following questions:

  1. What types of interventions are being delivered to prevent or treat adolescent de-pression and/or anxiety by promoting social inclusion?
  2. How effective are these interventions?
  3. Are there specific groups of adolescents for whom these interventions are most effective?
  4. What are the mechanisms through which these interventions reduce adolescent depression and anxiety?”

Our methods align with these questions. If you would like any specific information added, please let us know.

3)     The authors need to justify the eligibility criteria, perhaps this will allow more articles to be included in the review.

Thank you for highlighting this. We have included a detailed section in the revised draft to explain our selection of inclusion and exclusion criteria. We believe that these criteria are all justified, and hope that you find the expanded logic acceptable (see lines 297 onwards).

4)     The most important question that raises doubts and requires detailed justification: why were so many articles excluded? Is it possible that the original search query was formulated incorrectly or that the inclusion criteria were too strict?

The reason that so many studies were excluded was because we structured the initial search to be very, very broad. We did this because no systematic reviews of this kind have been done before, and we wanted to be absolutely sure that all possibly eligible publications were considered. In some reviews, the topic lends itself to narrow search terms. However, because social inclusion is quite a nebulous and intangible phenomenon, we had to ensure that our search would allow us to identify all possible types of interventions which would be eligible. Moreover, because of the age range in which we were interested, we could not refine our search to ‘children’ or ‘adults’, because age bracket definitions vary, and our age band cuts across these. So we again had to be over-inclusive in the search. For ease of reference, these were our terms:

Population. Adolescen* OR Child* OR Teen* OR Youth* OR Student* OR Pupil OR Learner OR Pube* AND

Intervention. Social inclusion OR social exclusion OR Social* exclu* OR social* inclu* OR acceptance OR sense of belonging OR anti-bullying OR school climate OR social acceptance OR social capital OR discrimination OR exclusion OR isolation OR participation OR social justice OR recreation OR Participat* OR integrat* OR Social network OR cohesion OR lonel* AND

Outcome. Anx* OR depress*

Yet, as much as our search needed to be broad, our inclusion and exclusion criteria needed to be rigorous and defensible. The combination of being thorough and rigorous has resulted in the large number of excluded studies. We do not believe that this diminishes the review.

However, we have added a lot of detail on this set of circumstances and our decisions to the manuscript, to support interpretation of our findings. Thank you for prompting us to do so.

Round 2

Reviewer 3 Report

As noted in my initial review, the manuscript reports a systematic review of the efficacy of interventions targeting social inclusion on depression and anxiety among adolescents. This is an interesting area of study, with potential implications on community-based interventions for depression and anxiety in adolescents. The authors are commended for providing a thorough response to the my and other reviewers' suggestios and a clear summary of the revisions. I have no further recommendations.

Author Response

Thank you so much for your response, and for your comments in the prior round. 

Reviewer 4 Report

Dear Authors,

Thank you very much for your attention to my recommendations and for the corrections made to the article.

I accept all answers and agree that the additions to the article made it better and more understandable for readers.

I understand that due to word count limitations, you cannot include the full search query in the text, however, it may be possible to include a description of the search terms you provide in your answer (see below)?

Population. Adolescen* OR Child* OR Teen* OR Youth* OR Student* OR Pupil OR Learner OR Pube* AND

Intervention. Social inclusion OR social exclusion OR Social* exclu* OR social* inclu* OR acceptance OR sense of belonging OR anti-bullying OR school climate OR social acceptance OR social capital OR discrimination OR exclusion OR isolation OR participation OR social justice OR recreation OR Participat* OR integrat* OR Social network OR cohesion OR lonel* AND

Outcome. Anx* OR depress*

Author Response

Thank you for this suggestion. We have now included a sentence (in red in the revised version) which summarises the search strategy. Thank you again for your extremely helpful comments on this piece.